# Liver Transcriptome Analysis Reveals a Potential Mechanism of Heat Stress Increasing Susceptibility to *Salmonella* Typhimurium in Chickens

**DOI:** 10.3390/biology14060720

**Published:** 2025-06-18

**Authors:** Qi Zhang, Yvqing Zhu, Zixuan Wang, Qinghe Li, Guiping Zhao, Qiao Wang

**Affiliations:** State Key Laboratory of Animal Nutrition, Institute of Animal Sciences, Chinese Academy of Agricultural Sciences, No2 Yuanmingyuan W Rd., Beijing 100193, China; zhangq0117@126.com (Q.Z.); 82101222364@caas.cn (Y.Z.); zixuanw2023@126.com (Z.W.); liqinghe@caas.cn (Q.L.); zhaoguiping@caas.cn (G.Z.)

**Keywords:** chicken, *Salmonella* Typhimurium, heat stress, RNA-seq, WGCNA

## Abstract

*Salmonella* infection causes severe disease in chickens and major economic losses for the poultry industry. Heat stress, which is becoming more common with global warming, makes chickens more vulnerable to infections, but how it does so remains incompletely understood. This study investigated how heat stress affects the ability of chickens to resist *Salmonella* infection. We compared healthy chickens to those infected with *Salmonella* Typhimurium, and to chickens exposed to both heat stress and *Salmonella* Typhimurium. The results showed that heat stress caused infected chickens to gain less weight, develop stronger inflammation, and experience higher mortality. Transcriptome analysis of the liver revealed that heat stress triggered excessive inflammatory responses and impaired antioxidant defenses. By integrating differential expression analysis, weighted gene co-expression network analysis, and cellular-level validation, we identified three key candidate genes (*PTGDS*, *SLC6A9*, and *WISP2*) that may play important roles in the host immune response during *Salmonella* Typhimurium infection under heat stress. These findings are important for disease-resistant poultry breeding and improving prevention strategies for poultry farming during hot weather.

## 1. Introduction

The poultry industry plays a vital role in providing high-quality and nutritious meat to meet the growing demands of the rapidly increasing global population [1]. However, beyond their nutritional and economic contributions, chickens serve as natural reservoirs for *Salmonella*, a major causative agent of human salmonellosis and one of the most significant foodborne diseases worldwide [2,3]. Industrialized and intensive farming practices, reduced trade barriers, and global warming collectively contribute to the increased prevalence and severity of *Salmonella* infections in poultry [4].

Climate change represents one of the greatest existential challenges to planetary and human health [5], with profound implications for agriculture [6]. Rising global temperatures have led to heat stress (HS), which adversely affects livestock health and has emerged as a critical concern for the global poultry industry [7]. Extensive research indicates that HS disrupts broiler physiology, leading to growth retardation, metabolic disorders, increased mortality, and immune imbalance [8,9]. Notably, HS has been shown to compromise avian immunity [10], thereby enhancing susceptibility to *Salmonella*. This effect may be linked to intestinal immune dysfunction, as *Salmonella* Enteritidis infection under HS has been reported to impair gut barrier integrity and promote inflammatory infiltration [11]. Furthermore, a recent study demonstrated that HS suppresses intestinal immune responses in *Salmonella* Typhimurium (ST)-infected broilers by downregulating the TLR4-NFκB-NLRP3 and TLR4-TBK1 signaling pathways [12]. Conversely, another study elucidated that the presence of pathogens or PAMPs under HS can result in cytokine storms and inflammatory cell death [13].

While it is evident that HS impairs broiler resistance to *Salmonella* infection, the underlying genetic mechanisms remain poorly understood. Transcriptomic analysis offers a powerful approach to elucidate the global gene expression profiles and complex biological processes controlling host immune responses. Therefore, this study aims to evaluate the phenotypic, physiological, and immunological responses of chickens to *Salmonella* infection under HS conditions, and to identify key genes and signaling pathways associated with disease resistance in the liver transcriptomes. Elucidating these mechanisms is essential for understanding the molecular basis of poultry immune responses to *Salmonella* infection under heat stress.

## 2. Materials and Methods

### 2.1. Experimental Population and Design

This study utilized Guang Ming (GM) broilers, a new white-feather broiler bred with the participation of the Institute of Animal Sciences, Chinese Academy of Agricultural Sciences (IAS-CAAS, Beijing, China). A total of 100 1-day-old chickens were obtained from the Changping Experimental Base of IAS-CAAS.

From 1 to 24 days of age, the chickens were maintained under the recommended environmental temperatures (33 ± 1 °C for days 1–7, 30 ± 1 °C for days 8–14, 26 ± 1 °C for days 15–21, and 22 ± 1 °C for days 22–24). At 25 days of age, the chickens were randomly divided into three groups: the control group (CTL group, n = 30), *Salmonella* Typhimurium-infected group (ST group, n = 30), and heat stress and *Salmonella* Typhimurium co-stimulation group (HS + ST group, n = 40). The chickens were then transferred to three separate isolation rooms (one room for the CTL group, one for the ST group, and one for the HS + ST group). Within their respective isolation rooms, all chickens were individually housed in cages. The environmental conditions (the temperature was controlled according to the design plan) within each isolation room were strictly controlled to minimize cross-contamination and environmental variability between groups. The CTL and ST groups were kept at 22 °C, while the HS + ST group was exposed to 33 °C for 8 days. On the 4th day of heat stress (at 28 days of age), the ST and HS + ST groups were orally challenged with 1 mL of PBS containing 2.5 × 10^10^ CFU ST, while the CTL group was given 1 mL of PBS orally. At 24 h post infection (hpi) (29 days of age), blood and livers were collected from 8 randomly selected chickens per group. The remaining chickens continued under their original conditions, and the survival rates were monitored throughout the study.

### 2.2. Bacteria Culture

*Salmonella enterica* serovar Typhimurium strain 21,484 (CICC) was cultured in Luria–Bertani (LB) broth at 37 °C with shaking (150 rpm) overnight, recovered, and then subcultured for an additional 12 h. The third-generation culture was used for infection experiments.

### 2.3. Weight Measurement

We measured the body weight of each group (CTL, ST, and HS + ST groups) at 8:00 a.m. at 25 days of age (before heat stress), 28 days of age (day 4 of heat stress, before *Salmonella* infection), and 29 days of age (1 day after *Salmonella* infection).

### 2.4. Phenotype Determination

Peripheral blood samples were collected from 8 chickens per group. Each sample was divided into two aliquots: one for blood smears and the other for serum isolation via centrifugation. After the blood smears were air-dried, they were stained with Wright–Giemsa. The heterophils (H), lymphocytes (L), and monocytes (M) were counted under an optical microscope at 100× magnification, with a total of 100 cells enumerated per sample [14]. The activities of the total antioxidant capacity (T-AOC) and total superoxide dismutase (SOD) serum levels were determined using commercial kits (Nanjing Jiancheng Bioengineering Institute, Nanjing, China). The concentrations of three inflammatory cytokines—interferon-γ (IFN-γ), interleukin-1β (IL-1β), and interleukin-8 (IL-8)—were measured in serum using the chicken enzyme-linked immunosorbent assay (ELISA) kit (Cusabio Biotech Co., Wuhan, China), following the manufacturer’s protocol.

### 2.5. Total RNA Isolation, cDNA Library Construction, and Sequencing

We extracted the total RNA from the liver samples using the QIAGEN RNeasy Kit (QIAGEN, Hilden, Germany) and removed the genomic DNA via the TIANGEN DNase Kit (TIANGEN Biotech, Beijing, China). The RNA purity, integrity, and concentration were checked with the Agilent Bioanalyzer 2100 System (RNA Nano 6000 assay kit, Agilent Technologies, Santa Clara, CA, USA).

For cDNA library construction, the purified mRNA was fragmented, and first-strand cDNA synthesis was initiated with 6-base random primers. Second-strand cDNA was generated using buffer, dNTPs, RNase H, and DNA polymerase I, followed by purification with the QIAQuick PCR kit (QIAGEN, Hilden, Germany) and elution in EB buffer. The double-stranded cDNA underwent end repair, A-tailing, adapter ligation, and size selection via agarose gel electrophoresis. Target fragments were PCR-amplified to finalize the library.

The libraries were quantified (Qubit 3.0, Thermo Fisher Scientific, Waltham, MA, USA), diluted to 1 ng/μL, and analyzed for insert size distribution (Agilent 2100 Bioanalyzer, Agilent Technologies, Santa Clara, CA, USA). The effective library concentration (>10 nM) was validated using the Bio-RAD CFX 96 system (Bio-Rad, Hercules, CA, USA) with the KIT-IQ SYBR GRN Q-PCR kit (Bio-Rad, Hercules, CA, USA). Qualified libraries were sequenced on the Illumina platform (Illumina Inc., San Diego, CA, USA) with a paired-end 150 bp (PE150) strategy.

### 2.6. Screening of Differentially Expressed Genes (DEGs) and Functional Enrichment Analysis

Read quality control was performed using FastQC v0.10.1 [15]. Adapters and low-quality reads were trimmed using Trimmomatic v0.39 [16]. Filtered reads were aligned to the Gallus gallus 6.0 reference genome using HISAT2 v2.1.0 [17]. Transcript abundance was quantified as FPKM (fragments per kilobase per million mapped reads) using RSEM v1.3.0 [18,19]. Differential expression analysis was conducted with DESeq2 v1.4.0 [20]. DEGs were screened with the threshold of |log2(Fold Change)| ≥ 1 and a *p*-value < 0.05. Gene expression values were provided in Appendix A. To investigate the function of DEGs, Gene Ontology (GO) and Kyoto Encyclopedia of Genes and Genomes (KEGG) pathway analyses were performed [21,22]. Terms or pathways with a *p*-value < 0.05 were regarded as significantly enriched.

### 2.7. Weighted Gene Co-Expression Network Analysis (WGCNA)

WGCNA was used to identify functional modules by constructing scale-free gene co-expression networks [23]. For preprocessing the data, low-expression genes (mean expression < 1) and outlier samples (based on abnormal branches in sample clustering dendrograms) were filtered. The Pearson correlation coefficient matrix between genes was calculated, and a suitable soft threshold *β* (based on a scale-free topology fitting index of *R*^2^ > 0.85) was selected to transform the correlation coefficients into an adjacency matrix using a power function. A topological overlap matrix (TOM) was subsequently computed to quantify gene co-expression similarity. Hierarchical clustering based on the TOM, combined with the dynamic tree-cutting algorithm, was used to delineate initial modules, which were merged based on correlations between module eigengenes. Finally, module eigengenes (MEs) were extracted, and their associations with target phenotypes were analyzed via Pearson correlation. Modules were selected for downstream analysis using thresholds of ∣correlation coefficient∣ ≥ 0.3∣ and a *p*-value < 0.05. Within the candidate modules, genes with |Module Membership| (|MM|) > 0.85 and |Gene Significance| (|GS|) >  0.30 were filtered as key genes. Gene interaction networks were exported using the exportNetworkToCytoscape function and visualized using Cytoscape v3.10.2 to identify the hub genes [24].

### 2.8. Cell Stimulation and Infection

HD11 cells (a chicken macrophage cell line) were seeded into culture plates and incubated at 37 °C with 5% CO_2_. The experiment included three groups: the CTL group (control group) and ST group (*Salmonella* Typhimurium-infected group) were kept at 37 °C, while the HS + ST group (heat stress and *Salmonella* Typhimurium co-stimulation group) was exposed to 41 °C for 3 days [25,26]. Subsequently, HD11 cells in the ST and HS + ST groups were infected with ST at 100 MOI. To promote bacterial–cell contact, the culture plates were centrifuged at 3000 rpm for 5 min, followed by continued incubation at the respective temperatures for 1 h to ensure bacterial invasion. Then, the cells were washed three times with PBS, and fresh medium containing 100 μg/mL gentamicin (Inalco Inc., San Luis Obispo, CA, USA) was added to continue culturing for 1 h. The cells were washed three times again with PBS to thoroughly remove extracellular bacteria. And the medium was replaced with fresh medium containing 10 μg/mL gentamicin and 10% FBS for 3 h [27]. Finally, cells were collected for subsequent experimental analysis.

### 2.9. Quantitative Real-Time PCR (qPCR) Analysis

The qPCR was performed using the ABI 7500 Real-Time PCR System (Thermo Fisher Scientific, Waltham, MA, USA) under the following conditions: for 10 min at 95 °C (initial denaturation), followed by 40 cycles of 95 °C for 15 s and 60 °C for 1 min. The relative mRNA expression was calculated using the comparative 2^−ΔΔCt^ method [28], with *β-actin* as the endogenous control. The primer sequences are detailed in Appendix A. Three biological replicates (with technical triplicates) were analyzed per condition.

### 2.10. Statistical Analysis

All statistical analyses were performed using GraphPad Prism v9.0.0. The data are presented as the mean ± standard deviation (mean ± SD). Intergroup comparisons were analyzed by one-way analysis of variance (ANOVA). When ANOVA indicated significance (*p*-value < 0.05), Tukey’s post hoc multiple comparison test was further applied. Significant differences in body weight data between groups are indicated by different lowercase letters (a, b; *p*-value < 0.05) in the tables. Significant differences for other phenotypic indicators between groups—including serum biochemical parameters (T-AOC and SOD), H/L ratio, serum cytokine concentrations (IFN-γ, IL-1β, and IL-8), and mRNA expression levels in HD11 cells—are denoted by asterisks in the figures (* *p*-value < 0.05 and ** *p*-value < 0.01).

## 3. Results

### 3.1. HS Exacerbated the Negative Impact of ST Infection on Weight Gain

Before exposure to HS (at 25 days of age), the initial body weights showed no significant differences among the groups (CTL, ST, and HS + ST group), indicating a balanced baseline body weight in the experimental animals (Table 1). After 3 days of HS (at 28 days of age), the body weight of the HS + ST group was significantly lower than that of both the ST and CTL groups (*p*-value < 0.05), while no significant difference existed between the ST and CTL groups, indicating that HS significantly suppressed weight gain in the HS + ST group. Following subsequent ST infection, at 24 hpi (at 29 days of age), the body weight of the ST group was slightly lower than the CTL group. The HS + ST group’s body weight remained significantly lower than that of the ST group (*p*-value < 0.05), and also showed a highly significant reduction compared to the CTL group (*p*-value < 0.01). In summary, HS significantly impeded animal growth and markedly exacerbated the negative impact of ST infection on weight gain.

### 3.2. HS Exacerbates Mortality in ST-Infected Chickens

To investigate the impact of HS on host responses to ST infection, we recorded the survival curves of the ST and HS + ST groups over a 5-day post-infection period. The results demonstrated that HS significantly enhanced the lethal effects of ST infection. The ST group showed a 95.45% survival rate, whereas the HS + ST group exhibited a markedly lower survival rate of 84.38% (Figure 1). These data indicated that HS substantially increases poultry susceptibility to ST infection.

### 3.3. HS Enhances Immune Factor Expression in ST-Infected Chickens

Assessment of HS markers revealed significantly elevated serum levels of T-AOC and SOD in the HS + ST group compared to both the CTL and ST groups (Figure 2A,B), confirming successful HS induction.

To examine ST-induced immune responses under HS, we analyzed the heterophil/lymphocyte (H/L) ratio and serum inflammatory cytokines. The H/L ratio, a well-established indicator for selecting ST-resistant chickens (lower ratios indicate higher resistance), progressively increased across groups (CTL < ST < HS + ST) (Figure 2C). These patterns suggested that HS may compromise host defenses by disrupting heterophil/lymphocyte homeostasis. At the serum inflammatory level, ST infection (ST group) upregulated IFN-γ, IL-1β, and IL-8 expression compared to in the CTL group. Notably, IFN-γ and IL-8 exhibited a stepwise increase (CTL < ST < HS + ST), with the HS + ST group displaying the peak expression (Figure 2D–F). These results clearly demonstrated that HS exacerbates inflammatory responses during ST infection.

### 3.4. Dynamic Regulation of DEGs in ST-Infected Chickens Under HS

To investigate the molecular mechanisms by which HS enhanced the host immune response to ST infection, we performed transcriptome sequencing on livers from the CTL, ST, and HS + ST groups. With screening thresholds of |log2(Fold Change)| ≥ 1 and *p*-value < 0.05, we identified DEGs in three comparison groups: ST vs. CTL, HS + ST vs. CTL, and HS + ST vs. ST (Figure 3A). Specifically, 329 DEGs were identified in ST vs. CTL (173 upregulated and 156 downregulated), 616 DEGs were identified in HS + ST vs. CTL (341 upregulated and 275 downregulated), and 305 DEGs were identified in HS + ST vs. ST (161 upregulated and 144 downregulated).

Gene expression analysis revealed gradient upregulation of inflammatory regulatory factors (*IL1R1*, *MAP3K9*), antimicrobial peptide gene (*AvBD9*), and prostaglandin synthesis gene (*PTGDS*) across the CTL, ST, and HS + ST groups. Conversely, tumor necrosis factor receptor (*TNFRSF13C*), WNT signaling gene (*WISP2*), and glycine transporter (*SLC6A9*) showed gradient downregulation. The chemokine (*CXCL13*), pattern recognition receptor (*TLR1A*), and superoxide dismutase (*SOD3*) peaked in the ST group but decreased in HS + ST group. Additionally, the pro-inflammatory factor *IL8L1* and heat shock proteins (*HSP90AA1*, *HSP25*) exhibited the highest expression levels in the HS + ST group (Figure 3B). Among these, 497 DEGs were specific to HS + ST vs. CTL, 210 DEGs were specific to ST vs. CTL, and 119 DEGs were shared (Figure 3C). We carried out GO (biological process) enrichment on the respective DEG sets. The specific genes of the ST vs. CTL comparison group were primarily enriched in immune response, leukocyte chemotaxis, and defense response to bacterium. The specific genes of the HS + ST vs. CTL comparison group were closely associated with oxygen transport, phospholipid homeostasis, and plasma membrane organization. And the shared genes of both comparison groups were significantly enriched in lipid transport and innate immune response (Figure 3D).

### 3.5. HS Activates Pro-Inflammatory Immune Pathways While Impairing Antioxidant Defenses in ST-Infected Chickens

We conducted GO and KEGG enrichment analysis on upregulated and downregulated DEGs in the HS + ST vs. ST comparison group.

GO term enrichment results (Figure 4A) showed that the upregulated DEGs were primarily enriched in heme binding, macrophage chemotaxis, the positive regulation of inflammatory response, positive regulation of apoptotic process, heat shock protein binding, and positive regulation of NF-κB transcription factor activity. The downregulated genes were mainly involved in the positive regulation of phagocytosis, phagocytosis recognition, and positive regulation of MAP kinase activity. These results indicated that HS enhanced inflammatory responses through the activation of the NF-κB signaling pathway, and induced the activation of genes related to heme binding and macrophage chemotaxis, which may explain the observed excessive inflammatory response in our experiments. Furthermore, the activation of heat shock protein binding pathways and the acceleration of apoptotic processes suggest that HS may participate in host immune responses by regulating programmed cell death pathways.

KEGG pathway enrichment results (Figure 4B) revealed that upregulated DEGs were mainly enriched in the NOD-like receptor signaling pathway and Toll-like receptor signaling pathway, while downregulated DEGs primarily participated in pentose and glucuronate interconversions and glutathione metabolism. Comprehensive analysis suggested that HS may enhance host immune responses to ST infection by coordinately activating multiple innate immune signaling pathways including the NLR, TLR, and mTOR signaling pathways. Conversely, inhibiting glutathione pathways suggested that HS impairs host antioxidant defenses, causing oxidative stress imbalance.

### 3.6. Weighted Gene Co-Expression Network Analysis and Identification of Key Module Genes

WGCNA was performed to identify modules correlated with the phenotype of the H/L ratio and serum inflammatory cytokine levels. Based on raw transcriptome data from 24 samples (8 CTL, 8 ST, and 8 HS + ST samples; 23,405 genes), we filtered stably expressed genes (SD ≤ 0.5), retaining 1702 genes for network construction. The scale-free topology fitting index reached 0.85, with the optimal soft threshold power set to 14 (Figure 5A). This threshold clustered the genes into 18 functional modules (the gray module containing unclustered genes was excluded from subsequent analysis) (Figure 5B).

Pearson correlation analysis evaluated correlations between module eigengenes and phenotypic traits, with |correlation coefficient| ≥ 0.3 and a *p*-value < 0.05 as significance thresholds. Notably, the yellow module (cor = 0.67, *p*-value < 0.001), blue module (cor = 0.50, *p*-value < 0.05), and purple module (cor = 0.54, *p*-value < 0.01) showed significant correlations with H/L ratio. The tan module (cor = −0.52, *p*-value < 0.01) demonstrated a significant correlation with the IFN-γ serum level. The turquoise module (cor = 0.44, *p*-value < 0.05) and green module (cor = −0.41, *p*-value < 0.05) were significantly correlated with the IL-8 serum level (Figure 5C).

With the thresholds of GS > 0.3 and MM > 0.8, we identified key genes associated with corresponding phenotypes in the significant modules (Appendix A). Analysis revealed that the blue module significantly correlated with the H/L ratio was enriched with multiple collagen genes (e.g., *COL1A1*, *COL8A1*, *COL3A1*, *COL1A2*, *COL4A2*, and *COL4A1*), primarily involved in tissue repair and fibrosis [29,30]. The purple module significantly associated with the H/L ratio was enriched with the lipid metabolism-related gene *APOA4*, potentially participating in cellular stress response and thereby profoundly influencing immune response processes [31]. In the tan module significantly correlated with IFN-γ, the key gene *CYP51A1* was identified, which participates in the biosynthesis of sterols (including cholesterol and potentially immunomodulatory sterols) [32], and whose metabolism is closely linked to immune cell functions (such as T-cell activation) and the regulation of inflammatory responses [33,34]. Among the modules significantly associated with IL-8, the turquoise module was enriched with the cytokine receptor gene *IL1R1*. As the primary receptor for the pro-inflammatory cytokine IL-1, the activation of IL1R1 constitutes a core pathway for initiating and amplifying inflammatory responses [35]. Meanwhile, within the green module, also associated with IL-8, the gene *SLC6A9* was enriched; this gene encodes glycine transporter 1 (GLYT1) and exhibits anti-inflammatory and cytoprotective effects [36]. These enriched key hub genes indicate their crucial roles in the combined stimulation of heat stress and *Salmonella* infection. Among all modules, the yellow module showed the strongest correlation (Figure 5C,D). We constructed a co-expression network for key genes in the yellow module and visualized it using Cytoscape (Figure 5E). This network included metallothionein genes (*MT4L* and *MT3*), inflammatory regulator (*IL1R2*), fatty acid transporter (*EXFABP*), growth factor (*FGF7*), prostaglandin synthase (*PTGDS*), and coagulation factor (*F13A1*). Notably, *MT4L* was identified as a potential hub gene.

### 3.7. Effect of PTGDS, WISP2, and SLC6A9 on Inflammatory Responses During ST Infection Under HS

By integrating differential expression analysis (HS + ST vs. ST) with WGCNA results, we identified three key hub genes: *PTGDS* (in the yellow module, significantly correlated with H/L ratio), *WISP2* (in the blue module, significantly correlated with H/L ratio), and *SLC6A9* (in the green module, significantly correlated with serum IL-8 levels), which showed differential expression in HS + ST vs. CTL comparisons.

To validate their functions, we established a HD11 cell stimulation model combining HS and ST infection. Experimental results demonstrated that ST significantly activated macrophage inflammatory responses, as evidenced by the markedly increased expression of inflammatory factors including *IL-1β*, *IL-8*, and *IFN-γ* (Figure 6A–C). Analysis of candidate gene expression patterns revealed that *PTGDS* was significantly upregulated in the ST group compared to the CTL group, and was further elevated in the HS + ST group (Figure 6D). In contrast, *WISP2* and *SLC6A9* exhibited suppressive expression in the ST group compared to the CTL group and remained downregulated in the HS + ST group (Figure 6E,F). Based on these distinct expression patterns, we propose that *PTGDS* acts as a positive regulator amplifying inflammatory responses, while *WISP2* and *SLC6A9* may participate in negative feedback mechanisms controlling excessive immune activation during ST infection and HS.

## 4. Discussion

Salmonellosis is a major infectious disease in the poultry industry, resulting in substantial economic losses [37,38]. Chickens’ resistance to *Salmonella* is influenced by multiple factors. Previous studies have reported increased splenic *Salmonella* colonization and intestinal inflammation in *Salmonella*-infected chickens under HS [11]. With the unequivocal trend of global warming, HS has emerged as a major environmental challenge in the poultry industry due to its adverse effects on health and production performance. Recent evidence has demonstrated that HS exacerbates infection-induced tissue damage, leading to severe pathological changes and elevated mortality [19,39]. However, the molecular mechanisms by which HS affects *Salmonella*-driven immune dysregulation remain incompletely understood. Through systematic phenotypic analysis, immune factor detection, and transcriptome analysis, this study provides a comprehensive elucidation of the molecular mechanisms underlying HS-aggravated ST infection in poultry. Our research provides novel theoretical perspectives for understanding the effects of environmental stress in pathogenic infections.

Our study revealed that HS alters host immune resilience through a dual mechanism: significantly exacerbating pro-inflammatory reactions while suppressing antioxidant defense capabilities. Experimental observations demonstrated decreased body weight gain, significantly higher mortality, and an abnormally elevated H/L ratio compared to the ST group. Extensive research has confirmed that the H/L ratio serves as a reliable and practical indicator for assessing poultry stress levels and health status [40,41,42,43]. Our previous studies established that chickens with lower H/L ratios exhibit stronger resistance to *Salmonella* [44]. These findings suggest that HS disrupts immune homeostasis by altering granulocyte proportions and impairs the host’s pathogen clearance capacity. Further investigation revealed that HS promoted a progressive upregulation of pro-inflammatory factors (IL-8, IFN-γ) in ST-infected chickens (CTL < ST < HS + ST). These expression patterns were correlated with the activation of NF-κB, TLR, and NLR pathways in liver transcriptomes, indicating that HS amplifies inflammatory phenotypes by enhancing innate immune signaling. These results align with previous poultry HS studies reporting increased infection susceptibility and excessive inflammatory responses. Both acute and chronic HS treatments significantly elevated serum inflammatory cytokine levels while increasing *Salmonella* load in host tissues [39]. Notably, the specific DEGs of the HS + ST group were predominantly enriched in oxygen transport and plasma membrane organization pathways, potentially reflecting HS-induced alterations in cell membrane permeability and oxygen radical accumulation. The concurrent activation of TLR/NLR pathways and suppression of glutathione metabolism suggested that the increased mortality of the HS + ST group is caused by the combined effects of hyperactivated innate immunity and compromised antioxidant defenses. Additionally, parallel studies in fish models demonstrated that HS increases pro-inflammatory cytokines, apoptosis regulators, and immune-related genes while disrupting the energy supply and oxidative metabolism [45,46]. Intriguingly, these fish studies suggested that prolonged and repeated heat exposure may conversely enhance HS recovery capacity—a discrepancy mainly attributable to fundamental differences in the thermoregulatory and immune systems between homeothermic and ectothermic animals.

The liver, as a pivotal immunometabolic hub, plays a central role in regulating gut barrier function and pathogen clearance. Through differential gene expression analysis and WGCNA network construction, this study successfully elucidated the molecular network characteristics of the synergistic effect of host responses to HS and ST infection. Our findings revealed that *PTGDS*, the hub gene of the yellow module (significantly correlated with the H/L ratio), showed consistent upregulation across the CTL, ST, and HS + ST groups, suggesting its pivotal role in the host immune response. The *PTGDS*-encoded prostaglandin D2 synthase (PGD2 synthase) catalyzes PGD2 production, exhibiting unique dual immunomodulatory properties. PGD2 can enhance macrophage chemotaxis to improve pathogen clearance [47]. Moreover, its metabolite 15d-PGJ2 might exacerbate tissue oxidative damage by activating pathways like NF-κB [39,48,49]. This dual function potentially explains why the HS + ST group demonstrated more severe inflammation and higher mortality compared with the ST group. Furthermore, excessive inflammatory factors (*IL8L1*) and prostaglandin mediators (such as PGD2, catalyzed by upregulated *PTGDS*) in the liver can directly act on intestinal tissues via portal circulation. These systemic inflammatory signals may disrupt intestinal epithelial tight junctions, increase intestinal mucosal permeability, create favorable conditions for *Salmonella* translocation and systemic dissemination, while further exacerbating local inflammatory damage in the gut. The WGCNA further identified significant downregulation of *SLC6A9* (glycine transporter) and *WISP2* (WNT signaling regulator) in ST-infected chickens under HS conditions, which may serve as a crucial regulator. Decreased expression of *SLC6A9* in the liver may block glycine utilization, disrupting glutathione synthesis and consequently weakening the host’s antioxidant defense capacity. This not only causes oxidative stress in the liver itself, but may also lead to a reduction in circulating antioxidant substances. Consequently, intestinal epithelial cells become more vulnerable to infection and reactive oxygen species (ROS) attacks induced by heat stress, further impairing barrier function [50,51,52]. Concurrently, *WISP2* downregulation may reduce its inhibitory effect on the WNT/β-catenin pathway, promoting excessive intestinal epithelial cell proliferation and compromising barrier integrity [53,54]. This process could both facilitate pathogen invasion and trigger inflammatory storms through abnormal immune cell activation, particularly through Th17 lymphocyte differentiation [55,56]. The dysregulation of these two pathways, combined with *PTGDS*-mediated inflammatory imbalance, forms an intricate cross-regulatory network that collectively drives a vicious cycle of “metabolic disorder–barrier damage–inflammation amplification”, ultimately leading to significantly increased host susceptibility to *Salmonella*.

## 5. Conclusions

This study elucidates the key pathways by which HS exacerbates ST infection in poultry. Our findings demonstrate that HS significantly compromises poultry disease resistance through inducing excessive inflammatory responses and disrupting antioxidant defense systems. Through transcriptome analysis and WGCNA network construction, we identified the key candidate genes (*PTGDS*, *SLC6A9*, and *WISP2*) that exhibited differential expression in the HS and ST co-exposed HD11 cell model, suggesting their pivotal regulatory roles in immune modulation. However, the precise mechanisms through which these genes mediate inflammatory responses require further investigation. Collectively, this work provides important insights and identifies critical molecular targets for enhancing disease resistance in poultry under heat stress conditions, offering valuable directions for future research in heat stress management and infectious disease control.

## Figures and Tables

**Figure 1 biology-14-00720-f001:**
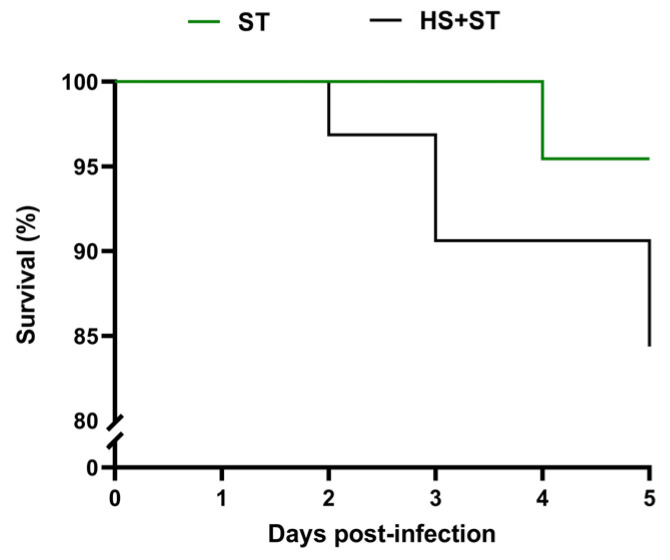
Survival curves for ST (upper curve, n = 22) and HS + ST (lower curve, n = 32) chickens during the first 5 days post infection.

**Figure 2 biology-14-00720-f002:**
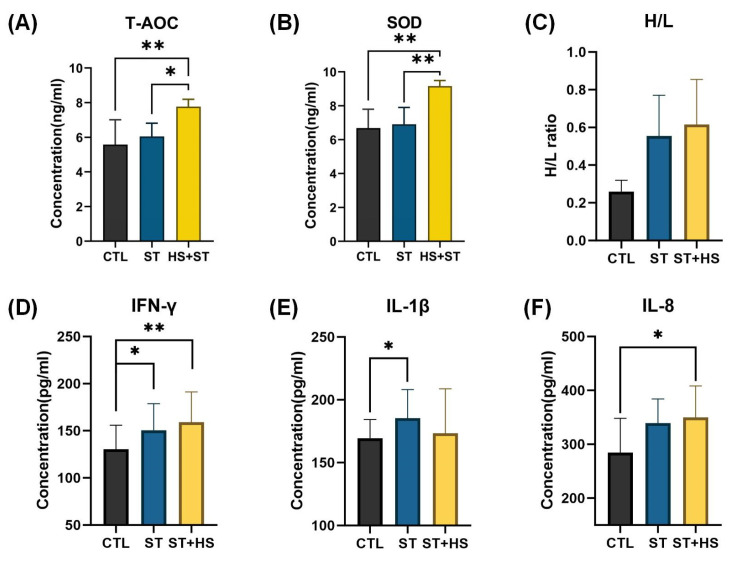
Phenotypic data in CTL (n = 8), ST (n = 8), and HS + ST (n = 8) groups. (**A**) Comparison of the total antioxidant capacity (T-AOC) blood serum concentration between CTL, ST, and HS + ST chickens. (**B**) Comparison of the superoxide dismutase (SOD) blood serum concentration between CTL, ST, and HS + ST chickens. (**C**) Comparison of the heterophil/lymphocyte (H/L) ratio in blood between CTL, ST, and HS + ST chickens. (**D**) Comparison of the interferon-γ (IFN-γ) blood serum concentration between CTL, ST, and HS + ST chickens. (**E**) Comparison of the interleukin-1β (IL-1β) blood serum concentration between CTL, ST, and HS + ST chickens. (**F**) Comparison of the interleukin-8 (IL-8) blood serum concentration between CTL, ST, and HS + ST chickens. * *p*-value < 0.05; ** *p*-value < 0.01.

**Figure 3 biology-14-00720-f003:**
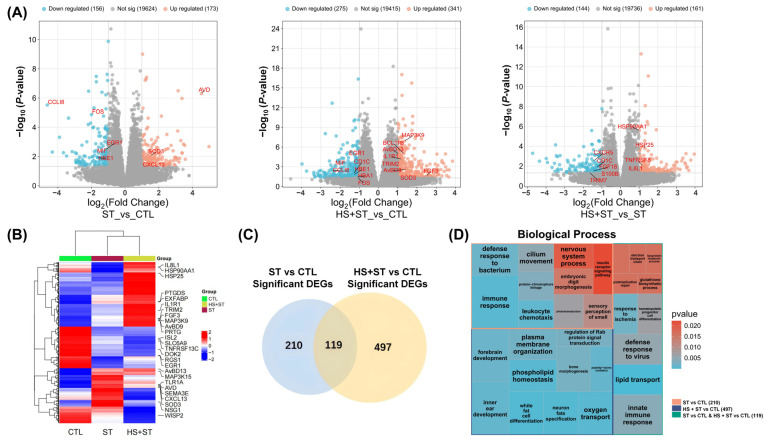
Liver transcriptome profiles in CTL (n = 8), ST (n = 8), and HS + ST (n = 8) groups. (**A**) Volcano plots of DEGs. Red spots represent the upregulated DEGs, and blue spots represent the downregulated DEGs. (**B**) Gene expression heatmaps in CTL, ST, and HS + ST groups. (**C**) Venn diagram of DEGs between “ST vs. CTL” and “HS + ST vs. CTL” comparison groups. (**D**) The GO (biological process) enrichment analysis of DEGs in three parts based on Venn diagram distribution. The red box represented the specific DEGs in “ST vs. CTL” comparison group, the blue box represented the specific DEGs in “HS + ST vs. CTL” comparison group, and the green box represented the common DEGs between “ST vs. CTL” and “HS + ST vs. CTL” comparison groups.

**Figure 4 biology-14-00720-f004:**
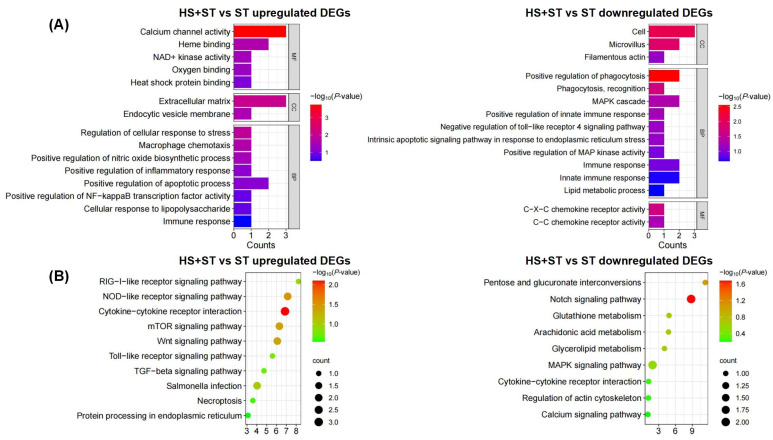
GO and KEGG enrichment analysis of upregulated and downregulated DEGs. (**A**) GO term analysis of upregulated DEGs (the left) and downregulated DEGs (the right). (**B**) KEGG signaling pathway enrichment analysis of upregulated DEGs (the left) and downregulated DEGs (the right).

**Figure 5 biology-14-00720-f005:**
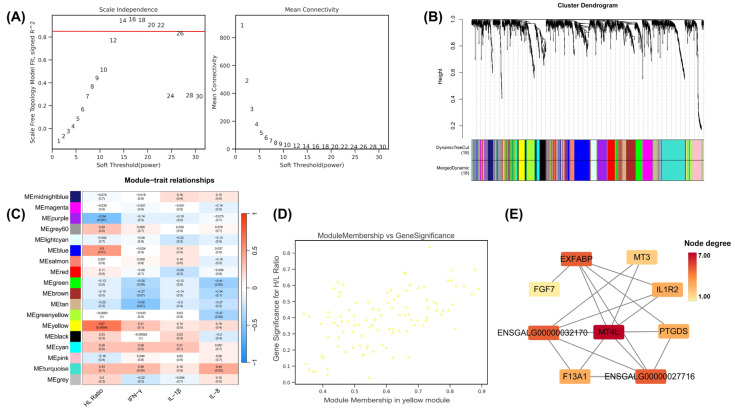
Implementation of WGCNA and identification of key module genes in CTL (n = 8), ST (n = 8), and HS + ST (n = 8) groups. (**A**) Analysis of network topology for various soft thresholding powers (1–30). (**B**) Clustering dendrogram of genes, with dissimilarity based on topological overlap, together with assigned module colors. (**C**) Module–trait associations. Each row corresponds to a module, and each column corresponds to a trait. Each cell contains the corresponding correlation and *p*-value. Color scale: Red represents positive correlation (max = 1), blue represents negative correlation (min = −1). (**D**) The scatterplot of the relevance between heterophil/lymphocyte (H/L) ratio and yellow module genes. (**E**) The regulatory network diagram of key genes in the yellow module. Node degree indicates the number of direct connections (edges) to other genes.

**Figure 6 biology-14-00720-f006:**
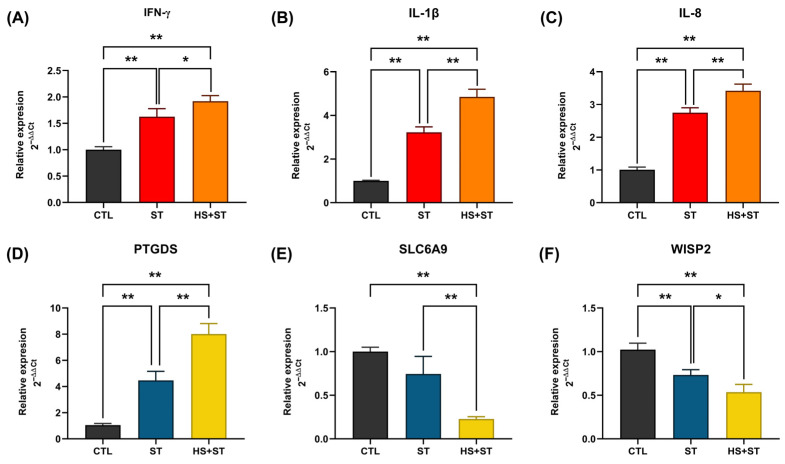
qPCR analysis of candidate gene expression was conducted in HD11 cells from CTL (n = 3), ST (n = 3), and HS + ST (n = 3) groups. (**A**) Interferon-γ (*IFN-γ*) mRNA expression. (**B**) Interleukin-1β (*IL-1β*) mRNA expression. (**C**) Interleukin-8 (*IL-8*) mRNA expression. (**D**) Prostaglandin D2 Synthase (*PTGDS*) mRNA expression. (**E**) Solute carrier family 6 member 9 (*SLC6A9*) mRNA expression. (**F**) WNT1-inducible signaling pathway protein 2 (*WISP2*) mRNA expression. * *p*-value < 0.05; ** *p*-value < 0.01.

**Table 1 biology-14-00720-t001:** The mean ± standard deviation for the effect of heat stress on the growth performance of *Salmonella* Typhimurium-infected chickens.

Parameters	Groups		
	CTL	ST	HS + ST
Before the start of heat exposure, 25 dpi	691.31 ± 121.79	688.64 ± 106.71	710.18 ± 73.45
One day after heat exposure, 28 dpi	1048.24 ± 156.74 ^a^	1048.47 ± 114.05 ^a^	936.24 ± 166.13 ^b^
One day after ST infection, 29 dpi	1115.74 ± 82.90 ^a^	1077.28 ± 113.78 ^a^	951.75 ± 140.60 ^b^

Note: Total sample size N = 100 (CTL: n = 30; ST: n = 30; HS + ST: n = 40). Within a row, means with different superscript letters (a, b) differ significantly at *p*-value < 0.05.

## Data Availability

All data supporting our findings are included in the manuscript. The Raw RNA-seq data can been downloaded in NCBI under the accession number PRJNA1178338 (Transcriptome data of chicken liver response to ST + HS) and are publicly accessible at https://www.ncbi.nlm.nih.gov/home/download/ (accessed on 27 October 2024).

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
