# Peer review of "Liver Transcriptome Analysis Reveals a Potential Mechanism of Heat Stress Increasing Susceptibility to Salmonella Typhimurium in Chickens"

_biology, 2025, doi:10.3390/biology14060720_

Round 1
Reviewer 1 Report
Comments and Suggestions for Authors
The research article, ‘Liver Transcriptome Analysis Reveals a Potential Mechanism of Heat Stress Increasing Susceptibility to Salmonella Typhimurium in Chickens’ has an important message for the scientific community and promises to attract a wide readership from fields such as animal nutrition, food safety, environmental science and public health. The subject matter is relevant and current. After careful review of the manuscript, some other comments I have are stated below, hoping that they will enhance the quality of the manuscript for acceptance and an interesting readership.
Abstract
- The study's objective is not well-written compared with what is presented in the introduction and should be revised. As it is, it lacks clarity and is misleading. A slight adjustment may bring out the exact information the authors want to pass on to the readers.
- Scientific names should be properly written throughout the text, including the title, abstract and keywords,
Introduction
- It is concise and contains adequate information such as background, knowledge gaps and the objectives of the study. However, the concluding remarks is wrongly placed and should be expunged. The findings of the study should be put in the results and the conclusion should summarise the impact of the study.
Materials and Methods
- What informed the decision on the size of experimental animals? Is that size rather not too small?
- How old were the animals when introduced into the study? The authors assumed we should know, but stating it simply does no harm. What was their average weight? Additional information regarding those may be necessary and may help in the design of similar future studies.
- Why was the growth performance across the experimental groups not reported?
- For adequate comparison and experimental design, one would have expected another experimental group of animals subjected to heat stress without Salmonella. I strongly believe that was necessary. Would the authors be kind to include that or provide a detailed explanation why they considered it unnecessary? Providing additional information may help to answer some questions potential readers may have.
- Could the authors also please provide a detailed explanation on why the groups had different numbers of experimental animals? Couldn’t that introduce bias? In my opinion equal number of animals should have been used to ensure that no bias is introduced. In that case, whatever is reported in the results will be logically attributed to the treatment effect.
- How were the isolation chambers arranged? Where were place placed? The reason for those questions is that it seems the treatments were not replicated. So, providing additional information on how the isolation chambers were arranged and where they were placed may make up for the lack of replication as it appears in the current form of the manuscript. I may be able to provide a favourable explanation for this, but the readers may not, bearing in mind that this article promises to attract readership from a wide audience with different scientific backgrounds.
- Adequate and relevant references should be cited where necessary to give credence to the adopted methodology and give credit to those to whom it is due.
- The design of the study looks like a completely randomized design and it was expected that the data would have been analysed in line with one-way analysis of variance. Could the authors provide additional information to justify the use of a two-tailed Student t-test?
Results
- This section is well-written. The findings were adequately presented.
Discussion
- This section is also considered adequate. The findings presented in the result section were justified, explained and adequately compared with findings from the previous similar studies, bolstered with relevant references.
Conclusion
- The conclusion was drawn from the findings of the study and it is considered adequate, summarizing the impacts of the study. However, the claim that the ‘study reveals the molecular mechanisms by which heat HS exacerbates ST infection’ is too broad and should be narrowed down. The study doesn’t seem to have the required evidence to back that up. Hence, the ambitious claims in the conclusion should be narrowed down. Moreover, the authors stated that ‘The findings provide theoretical foundations…’ To avoid contradiction, the suggested amendment may be necessary. Borrowing an idea from the proposed title may help to resolve that.
Author Response
Thank you very much for taking the time to review this manuscript. Please find the detailed responses below and the corresponding revisions in the re-submitted files.
Comments 1: The study's objective is not well-written compared with what is presented in the introduction and should be revised. As it is, it lacks clarity and is misleading. A slight adjustment may bring out the exact information the authors want to pass on to the readers.
Response 1: Thank you for your suggestion. We have made revisions to the abstract section to make it clearer; see lines 28-48 in the revised manuscript.
Comments 2: Scientific names should be properly written throughout the text, including the title, abstract and keywords.
Response 2: Thank you for your suggestion. We have corrected the formatting of scientific names “Salmonella Typhimurium” in the revised manuscript: at lines 31, 38, 45 and 515. And change “Salmonella enterica serovar Typhimurium strain” to “Salmonella enterica serovar Typhimurium strain” at line 111.
Comments 3: It is concise and contains adequate information such as background, knowledge gaps and the objectives of the study. However, the concluding remarks is wrongly placed and should be expunged. The findings of the study should be put in the results and the conclusion should summarise the impact of the study.
Response 3: Thank you for your suggestion. We have revised the introduction section and added a summary of the research implications at lines 79-81.
Comments 4: What informed the decision on the size of experimental animals? Is that size rather not too small?
Response 4: We sincerely appreciate your thorough review of our research, particularly the valuable feedback regarding the sample size in the animal experiments. We fully recognize that appropriately increasing the sample size would enhance the reliability of the experimental results and strengthen the persuasiveness of the conclusions. However, we also confirm that the current sample size is scientifically justified. We hereby provide a detailed explanation of the rationale for determining the sample size in this study (total of 100 chickens, grouped as: Control group: 30 chickens, Salmonella infection group: 30 chickens, Combined heat stress and Salmonella infection challenge group: 40 chickens). Our determination was based on referencing relevant existing studies and carefully balancing scientific rigor with the animal ethics principles of the "3Rs" (Reduction, Refinement, Replacement). We examined the experimental designs in published literature relevant to this study (heat stress, Salmonella infection, and their interaction). One study investigating heat stress in chickens employed 72 chickens randomly divided into control, chronic heat stress, and acute heat stress groups (n=24 per group), revealing differences between chronic and acute stress effects [1]. Another study exploring the combined effects of heat stress and Salmonella infection in poultry used 40 broilers divided into control, HS group, ST group, and HS+ST group (n=10 per group) to investigate the impact of high-temperature environments on poultry immune status [2]. These studies were peer-reviewed and accepted, successfully detecting significant effects. Based on these robust precedents, our sample size setting (≥30 chickens per group) exceeds those of the referenced studies. Furthermore, considering animal welfare and strictly adhering to ethical requirements while meeting the scientific objectives, we ultimately determined the current experimental scale.
References:
[1] Alhenaky A, Abdelqader A, Abuajamieh M, Al-Fataftah AR. The effect of heat stress on intestinal integrity and Salmonella invasion in broiler birds. J Therm Biol. 2017;70(Pt B):9-14. doi: 10.1016/j.jtherbio.2017.10.015.
[2] Tang LP, Li WH, Liu YL, Lun JC, He YM. Heat stress inhibits expression of the cytokines, and NF-κB-NLRP3 signaling pathway in broiler chickens infected with salmonella typhimurium. J Therm Biol. 2021;98:102945. doi: 10.1016/j.jtherbio.
Comments 5: How old were the animals when introduced into the study? The authors assumed we should know, but stating it simply does no harm. What was their average weight? Additional information regarding those may be necessary and may help in the design of similar future studies.
Response 5: Thank you for your suggestions. All chicks arrived at the experimental facility at 1-day-old. We have added this information at line 90 of the revised manuscript.
The "average body weight of the animals" you pointed out is also a key piece of information. Therefore, we have compiled this data and added it to the main text: We weighed the body weights of each group before exposure to heat stress (at 25 days of age), after 3 days of heat stress (before Salmonella infection, at 28 days of age), and one day after Salmonella infection (at 29 days of age). This content has been added in the revised manuscript (lines 38, 115-118, 211-227, 420-422 of the revised manuscript).
Comments 6: Why was the growth performance across the experimental groups not reported?
Response 6: Thank you for your suggestion. We have added the effects of different treatments on the growth performance (body weight) of each group in the Results section (lines 211-227 of the revised manuscript).
Comments 7: For adequate comparison and experimental design, one would have expected another experimental group of animals subjected to heat stress without Salmonella. I strongly believe that was necessary. Would the authors be kind to include that or provide a detailed explanation why they considered it unnecessary? Providing additional information may help to answer some questions potential readers may have.
Response 7: Thank you for your suggestion. We would like to explain here why we did not join a separate heat stress group:
The core strategy of our experimental design is to maximize the analysis of the impact of heat stress on the host's infection response by strictly controlling variables (temperature, infection status) and utilizing a direct comparison between the HS+ST group (heat stress + Salmonella Typhimurium infection) and the ST group (normal temperature + Salmonella Typhimurium infection). The primary purpose of designing these three groups is as follows:
- HS+ST Group(Heat Stress + SalmonellaTyphimurium Infection): This represents the core experimental group we focus on, capturing the state where the target stressor (heat stress) coexists with the key challenge (Salmonella Typhimurium infection).
- ST Group(Normal Temperature + SalmonellaTyphimurium Infection): This serves as the crucial infection control group, providing the baseline response of the host to Salmonella Typhimurium infection under "normal" ambient temperature conditions.
- CTL Group(Normal Temperature + Non-infected): This provides a reference for the basal physiological state.
While adding an HS group (Heat Stress + Non-infected) could offer more comprehensive background information on the physiological effects of heat stress alone, we are concerned that it might divert attention from the core objective of this study and significantly increase the complexity and resource demands of the research.
We believe the current design effectively and robustly addresses our core scientific question. Furthermore, we strengthen the reliability of the results through appropriate literature citations and thorough discussion.
Comments 8: Could the authors also please provide a detailed explanation on why the groups had different numbers of experimental animals? Couldn’t that introduce bias? In my opinion equal number of animals should have been used to ensure that no bias is introduced. In that case, whatever is reported in the results will be logically attributed to the treatment effect.
Response 8: Thank you for your suggestion. The question you raised regarding the difference in the initial number of experimental animals across the groups (30 in the CTL group, 30 in the ST group, and 40 in the HS+ST group) and its potential for introducing bias is highly important. This design was based on clear scientific considerations aimed at ensuring the reliability and validity of the results, rather than introducing bias. Below is a detailed explanation:
The HS+ST group (heat stress + Salmonella Typhimurium infection combined stimulation) is the key treatment group in this study, subjected to the combined pressure of both heat stress and pathogenic infection. Based on preliminary research experience and literature reports, this dual stress is highly likely to result in a higher mortality rate or a greater incidence of individuals deteriorating in condition before sampling compared to the single stress (ST group) or the control group (CTL group).
Setting the initial number of the HS+ST group to 40 was primarily to provide a "safety buffer," preventing insufficient sample size in this group due to unexpected attrition. We anticipated that this group might have a higher risk of individual loss during the experiment (from the start of heat treatment until the 24-hour sampling point). Increasing the initial number ensured that at the planned sampling time point (24 hours post-infection), we would still have a sufficient number (targeting 8 individuals) of evaluable individuals meeting the sampling criteria for key phenotypic assays (blood smears, serum cytokines).
Although the initial numbers differed, the main critical data collection (blood smears, serum cytokines, liver transcriptome) and analysis were based on randomly selected subsets of individuals from each group at the identical time point (24 hours post-Salmonella infection), with completely equal numbers per group for analysis. The strict randomization process meant that every chicken had an equal chance of being selected, independent of any subjective judgment by the researchers, effectively eliminating potential selection bias arising from the initial group size differences. The observed differences between groups reliably reflect the true effects of the different treatments (control, single infection, dual stress) on the host immune response.
Furthermore, during the experimental design phase, we consulted published studies with similar methodologies. For instance, in one investigation on Salmonella Enteritidis infection [3], researchers randomly allocated 100 animals to the uninfected group and 200 to the Salmonella Enteritidis-infected group based on experimental requirements and anticipated attrition rates. This study successfully conducted a 7-day survival curve analysis following Salmonella infection. At 1, 3, 7, and 21 days post-infection (dpi), 20 chickens were randomly selected from each experimental group for sampling to assess bacterial loads in various tissues, serum cytokine concentrations, and other parameters. The research also comprehensively examined the impact of heterophil/lymphocyte (H/L) ratios on disease resistance against Salmonella. Both the experimental design—including its differential group sizes—and its findings have undergone peer review and been published in an academic journal. This established precedent provides reference and validation for our approach of assigning a relatively larger initial sample size to the HS+ST treatment group.
References:
[3] Thiam M, Barreto Sánchez AL, Zhang J, Wen J, Zhao G, Wang Q. Investigation of the Potential of Heterophil/Lymphocyte Ratio as a Biomarker to Predict Colonization Resistance and Inflammatory Response to Salmonella enteritidis Infection in Chicken. Pathogens. 2022; 11(1):72. doi: 10.3390/pathogens11010072.
Comments 9: How were the isolation chambers arranged? Where were place placed? The reason for those questions is that it seems the treatments were not replicated. So, providing additional information on how the isolation chambers were arranged and where they were placed may make up for the lack of replication as it appears in the current form of the manuscript. I may be able to provide a favourable explanation for this, but the readers may not, bearing in mind that this article promises to attract readership from a wide audience with different scientific backgrounds.
Response 9: Thank you for your suggestion. We have added specific details on the arrangement of animal grouping in the isolation room in lines 97-103 of the materials and methods section.
Comments 10: Adequate and relevant references should be cited where necessary to give credence to the adopted methodology and give credit to those to whom it is due.
Response 10: Thank you for your suggestion. We have added relevant literature references (No. 14, 21-27, 29-36) to the method used in the revised manuscript (lines 125, 158, 162, 177 183, 190, 345, 338, 351, 353, 356, 359).
Comments 11: The design of the study looks like a completely randomized design and it was expected that the data would have been analysed in line with one-way analysis of variance. Could the authors provide additional information to justify the use of a two-tailed Student t-test?
Response 11: We sincerely appreciate your thorough and meticulous review of our manuscript and the valuable feedback you provided. Your correction regarding the statistical analysis method (specifically, that analysis of variance should be used for three-group comparisons instead of the two-tailed t-test) is highly pertinent and constructive, and we fully concur.
In the initial analysis, for data involving comparisons among the three groups (CTL, ST, HS+ST) – such as serum biochemical parameters, H/L ratio, cytokine concentrations, qPCR results, etc. – we employed only two-tailed Student's t-tests for pairwise comparisons. This approach did not adequately account for the experimental design, which is a one-factor completely randomized design comprising three independent treatment groups. Relying solely on t-tests for multiple pairwise comparisons increases the risk of Type I error (false positives).
Following your recommendation, we have re-analyzed all data involving three-group comparisons, replacing the t-tests with one-way analysis of variance (One-way ANOVA) combined with Tukey's Honest Significant Difference (HSD) test for post-hoc pairwise comparisons. A detailed description of the revised statistical methods has been added to the revised manuscript at lines 199-209. Figures updated with the new significance markers include: Fig. 2E-F, 6A, 6C, 6E-F.
Comments 12: The conclusion was drawn from the findings of the study and it is considered adequate, summarizing the impacts of the study. However, the claim that the ‘study reveals the molecular mechanisms by which heat HS exacerbates ST infection’ is too broad and should be narrowed down. The study doesn’t seem to have the required evidence to back that up. Hence, the ambitious claims in the conclusion should be narrowed down. Moreover, the authors stated that ‘The findings provide theoretical foundations…’ To avoid contradiction, the suggested amendment may be necessary. Borrowing an idea from the proposed title may help to resolve that.
Response 12: Thank you for your suggestion. We have made corresponding modifications to this section in lines 485-486, 489, and 493-496 of the revised manuscript.
Additional clarifications:
- We have added the Simple Summary in the revised manuscript (lines 13-27).
- We have added the approval number (IAS2021-31) to our Ethics Statement (line 86).
- We have rechecked the funding for this study and corrected it to “This study was supported by grants from the National Key Research and Development Program (2022YFF1000203), the Biological Breeding-National Science and Technology Major Project (2023ZD0405302) and the Central Public-Interest Scientific Institution Basal Research Fund (No. 2023-YWF-ZYSQ-07)” (line502-505).
- We have added new full names and their abbreviations in the abbreviation table (line 515).
Reviewer 2 Report
Comments and Suggestions for Authors
This manuscript presents a well-executed in vivo and in vitro study investigating how heat stress (HS) exacerbates susceptibility to Salmonella Typhimurium (ST) infection in chickens. Through the integration of physiological assessments, liver transcriptomic profiling, WGCNA, and macrophage cell validation, the authors identify three hub genes—PTGDS, WISP2, and SLC6A9—that may mediate this interaction.
The study is relevant, methodologically sound, and provides novel insight into the molecular crosstalk between environmental stress and host-pathogen interaction. The findings have potential implications for poultry breeding and HS mitigation strategies under climate change scenarios.
While the authors appropriately connect transcriptomic signatures to phenotypes (H/L ratio, cytokines), more discussion on how systemic liver responses influence gut barrier function and pathogen clearance would enhance interpretation.
Figure legends should include sample sizes and full gene names where relevant.
Supplementary tables (especially Table S3 listing key WGCNA genes) should be better integrated into the main results narrative.
Comments on the Quality of English LanguageGenerally clear, but a few grammatical and stylistic revisions are needed. Examples:
“significantly exacerbated Salmonella typhimurium-induced inflammatory phenotypes” consider simplifying.
“Conversely, the inhibition of glutathione metabolism pathways indicated that HS may impair antioxidant defense…” more concise phrasing would help.
Author Response
Thank you very much for taking the time to review this manuscript. Please find the detailed responses below and the corresponding revisions in the re-submitted files.
Comments 1: While the authors appropriately connect transcriptomic signatures to phenotypes (H/L ratio, cytokines), more discussion on how systemic liver responses influence gut barrier function and pathogen clearance would enhance interpretation.
Response 1: Thank you for your suggestion. To supplement the discussion on “how systemic liver responses influence gut barrier function and pathogen clearance would enhance interpretation” we have added relevant content in the revised manuscript: at lines 448-449, 460-466, and 469, 471-474.
Comments 2: Figure legends should include sample sizes and full gene names where relevant.
Response 2: Thank you for your suggestions. We have added the sample sizes and full gene names to the figure legends, with specific modification locations detailed on lines 226–227, 236, 253–260, 288, 368–369, 374, and 396–401 of the revised manuscript. Additionally, during checking the sample size, we discovered that the 8 samples per group already used for sampling were not excluded from the survival rate calculation (originally calculated as ST: n=30, HS+ST: n=40). Consequently, we have corrected the survival rates (ST: n=22, HS+ST: n=32) and replotted Figure 1 (Fig. 1) accordingly. These adjustments appear on lines 232–233 of the revised manuscript.
Comments 3: Supplementary tables (especially Table S3 listing key WGCNA genes) should be better integrated into the main results narrative.
Response 3: Thank you for your suggestion. We have supplemented the results analysis for this part at lines 343-360 of the revised manuscript.
Comments 4: Generally clear, but a few grammatical and stylistic revisions are needed. Examples:
“significantly exacerbated Salmonella typhimurium-induced inflammatory phenotypes” consider simplifying.
“Conversely, the inhibition of glutathione metabolism pathways indicated that HS may impair antioxidant defense…” more concise phrasing would help.
Response 4: Thank you for your suggestion. We have simplified these sentences; see lines 38-39 and 318-319 in the revised manuscript.
Additional clarifications:
- We have added the Simple Summary in the revised manuscript (lines 13-27).
- We have added the approval number (IAS2021-31) to our Ethics Statement (line 86).
- We have rechecked the funding for this study and corrected it to “This study was supported by grants from the National Key Research and Development Program (2022YFF1000203), the Biological Breeding-National Science and Technology Major Project (2023ZD0405302) and the Central Public-Interest Scientific Institution Basal Research Fund (No. 2023-YWF-ZYSQ-07)” (line502-505).

Round 2
Reviewer 1 Report
Comments and Suggestions for Authors
I appreciate the authors' willingness to revise the manuscript as suggested. After a careful revision of the revised version of the manuscript and the authors' response, I recommend the article for acceptance.